# Child Maltreatment and Public Health: Do Gaps in Response during the COVID-19 Pandemic Highlight Jurisdictional Complexities?

**DOI:** 10.3390/ijerph18136851

**Published:** 2021-06-25

**Authors:** Colleen M. Davison, Susan J. Thanabalasingam, Eva M. Purkey, Imaan Bayoumi

**Affiliations:** 1Department of Public Health Sciences, Queen’s University, Kingston, ON K7L 3N6, Canada; 2Department of Medicine, Queen’s University, Kingston, ON K7L 3N6, Canada; tsusan@qmed.ca; 3Department of Family Medicine, Queen’s University, Kingston, ON K7L 3N6, Canada; eva.purkey@queensu.ca (E.M.P.); bayoumi@queensu.ca (I.B.)

**Keywords:** child maltreatment, neglect, child abuse, child welfare, child protection

## Abstract

*Objective*: Countermeasures introduced during the COVID-19 pandemic produced an environment that placed some children at increased risk of maltreatment at the same time as there were decreased opportunities for identifying and reporting abuse. Unfortunately, coordinated government responses to address child protection since the start of the pandemic have been limited in Canada. As an exploratory study to examine the potential academic evidence base and location of expertise that could have been used to inform COVID-19 pandemic response, we undertook a review of child maltreatment research across three prominent Canadian professional journals in social work, medicine and public health. *Methods*: We conducted a pre-pandemic, thirteen-year (2006–2019) archival analysis of all articles published in the Canadian Social Work Review (CSWR), the Canadian Medical Association Journal (CMAJ) and the Canadian Journal of Public Health (CJPH) and identified the research articles that related directly to child maltreatment, child protection or the child welfare system in Canada. *Results*: Of 11,824 articles published across the three journals, 20 research papers relating to child maltreatment, child protection or the child welfare system were identified (CJPH = 7; CMAJ = 3; CSWR = 10). There was no obvious pattern in article topics by discipline. *Discussion*: Taking these three prominent professional journals as a portal into research in these disciplines, we highlight the potential low volume of academic child maltreatment research despite the importance of the topic and irrespective of discipline. We believe that urgent transdisciplinary collaboration and overall awareness raising for child protection is called for at the time of the COVID-19 pandemic as well as beyond in Canada.

## 1. Introduction

In response to the COVID-19 pandemic, urgent public health countermeasures were implemented worldwide, including closures of schools and workplaces, the suspension of non-essential public services, stay at home measures and mandated physical distancing. Families have faced increasing financial and social pressures, dramatic changes to daily life, increased social isolation and decreased availability of some health and social services [1]. In this context, alarm bells have been ringing across the globe warning of potential increases in child abuse and maltreatment [2]. Indeed, the COVID-19 pandemic appears to represent a perilous situation for vulnerable children. Mandatory lockdowns and mandated physical distancing have meant children at risk of maltreatment have been isolated with increasingly stressed and potentially abusive parents [1]. Unprecedented increases in child screen time, and the potential for unsupervised internet usage, may have led to increased risk of online predators accessing children [3]. The closure of schools, workplaces and social services has led to fewer public and social interactions across our communities. Job losses, and the deteriorating financial situations of families leading to increased stress on caregivers, have been linked in the past to increased incidence of child maltreatment [4]. Jurisdictions across the globe are showing dramatic reductions in child maltreatment reports [5] at the same time as direct calls from children to emergency help lines are up [6]. UN Special Rapporteur on the Sale and Sexual Exploitation of Children, MF Singhateh, has stated that globally, *“The damage to millions of children will be devastating if we are slow in mobilising child protection services for early detection and prevention”* [7].

Child maltreatment constitutes all forms of physical, emotional or sexual abuse, as well as neglect or exploitation, resulting in actual or potential harm to the child’s health, survival, development or dignity [8]. Exposure to intimate partner violence can also constitute a form of child abuse or maltreatment. Not only is child abuse a societal issue of broad concern, the effects of child maltreatment are significant for individuals in the short and long term. Maltreated children are at greater risk of suffering physical and mental health concerns in the immediate term, as well as facing longer-term risks of adult morbidity and premature death [9,10]. While many governments have acted to mitigate some economic impacts of the COVID-19 pandemic, coordinated actions to mitigate the impacts of the pandemic on family conflict and violence, child abuse, and deteriorating mental health have been late to emerge. Likewise, the discussion about reopening schools was a secondary priority to economic recovery, despite the significant impact of prolonged school closure on child well-being and potential exposure to maltreatment [11,12]. There has been a growing swell of child maltreatment warnings but, as yet, a near vacuum of empirical study or coordinated response. If the countermeasures put children at higher risk for maltreatment, and if this abuse has serious short- and long-term implications, why were mitigation plans not being more greatly prioritized at the outset? 

In 2011, the Canadian Journal of Public Health published an article by Tracie O. Afifi titled: Child Maltreatment in Canada: An understudied public health problem (volume 102) [13]. In this commentary, Afifi notes that high-quality studies on child maltreatment issues are critical for informing interventions and that compared to other parts of the world: “the number of studies conducted in Canada is far fewer and the data used to study this important public health problem are less diverse” [13], p. 459. We believe that part of this problem arises because of the complexities of child maltreatment research [14] and also because of the mixed jurisdiction of child protection in Canada and the low priority it has often been shown in policy decisions.

Child protection in Canada is the combined responsibility of many stakeholders. Formal child protection services are within the realms of social work and law enforcement and, historically, children’s aid societies and child and family services have been developed largely independently of health care or public health. We believe the lack of comprehensive child protection mitigation plans in the context of the COVID-19 pandemic may be evidence of the implications of this historical divide in authority or jurisdiction. To explore this hypothesis further, we decided to conduct an archival analysis of journal articles published in three prominent Canadian professional journals (the Canadian Journal of Public Health, the Canadian Medical Association Journal and the Canadian Social Work Review) for more than a decade pre-pandemic. We aimed to determine the extent and nature of child maltreatment research being published in these venues and to see where “ownership” or jurisdiction of child protection issues may lie from an academic standpoint. This was an exploratory study recognizing there are many journals we could have used as examples.

## 2. Materials and Methods

The titles of all articles published in the Canadian Journal of Public Health (CJPH), the Canadian Medical Association Journal (CMAJ) and the Canadian Social Work Review (CSWR) between January 2006 and December 2019 were screened for relevance to children, child health, the child welfare system, parents and/or families. Abstracts for all articles that passed the initial title screen were read. Articles relevant to any aspect of “child maltreatment” (also included child abuse, neglect, maltreatment) or “the child welfare system” were noted. Reasons for any exclusions were noted. Full-text articles were screened for any title that did not have an abstract. The full texts of all included articles were obtained and reviewed. Articles that represented research publications (as opposed to commentaries, letters, opinion pieces, etc.) were noted. Figure 1 is the search flow chart.

## 3. Results

In total, 1729 articles were published in the CJPH from 2006 to 2019, of which 21.3% pertained to children or families with children. For the CMAJ, in the same time period, 9811 articles were published, of which 6.0% pertained to children or families. The CSWR published 284 articles, of which 15.8% pertained to children or families. After abstract and full-text review, 13 articles from the CJPH, 20 from the CMAJ and 20 from the CSWR were found to be specifically related to child maltreatment or the child welfare system, for a total of 53 articles across the three journals over the thirteen-year period between 2006 and 2019. Among these relevant articles, 7 articles in the CJPH, 3 in the CMAJ and 10 in the CSWR were research reports (see Table 1 for information about these articles). Thus, 11,824 articles in total were published in these three journals between 2006 and the end of 2019, of which there were only 20 research papers pertaining directly to child maltreatment or the child welfare system (~0.2%). A qualitative review of the titles and abstracts of the 20 research studies indicates that, although we might have expected otherwise, there was no obvious pattern in the topics or focus of the research evident in these studies across the three journals.

## 4. Discussion

This was an exploratory study of the extent and nature of the child maltreatment academic research evidence base in three prominent Canadian journals in social work, medicine and public health. This is evidence and expertise that could have been drawn on to inform aspects of the COVID-19 pandemic in Canada. We undertook this review as a way to explore disciplinary ownership, jurisdiction and interest in child maltreatment issues across these disciplines. The results indicate very limited academic research on child protection issues overall in these venues. Only 20 relevant research articles were published out of a possible 11,824 papers in these journals over this period. Using these three journals as a portal, there appears to be a relatively limited number of researchers or research teams undertaking academic research related to child maltreatment issues in Canada. This represents a potentially limited peer-reviewed evidence base from which to inform policy and decision making. Limitations in academic research may be reflections of the level of research interest, prioritization, capacity, feelings of jurisdiction, data sources, funding, etc. Potter and colleagues noted that there are few who have the experience and interest to draw from existing data sources related to child maltreatment in Canada [35]. There is also a small number of people or organizations who are creating new quality sources of data [14]. The convergence of these existing research limitations and the realities of the COVID-19 pandemic, with a very likely increase in child maltreatment and recent reductions in support and monitoring for children [36], is gravely concerning. While there are very important contributions, including work in the PreVail and Vega projects [https://vegaproject.mcmaster.ca, accessed on 16 June 2021], which have not been published in the CJPH, the CMAJ, or the CSWR, as Afifi’s commentary [13] from 2011 noted, we believe Canadian child protection literature is still very limited. This was certainly not expected, particularly in a social work journal. In the present context of a potential child maltreatment and COVID-19 “syndemic” [37], researchers who might have expertise and interest to advocate for greater priority towards these concerns and the existence of evidence to inform effective interventions across social work, medicine and public health are essential. 

Canada is a supporter of the UN Sustainable Development Goals and requires data pertaining to its peace and security targets (including violence against children). Canada is also a signatory to the United National Convention on the Rights of the Child. Article 19 specifically states that all children have the right to be protected from abuse and neglect. Governments require evidence to track action and progress on these targets. They are also accountable to the children and families who make up their constituents. The lack of child protection mitigation plans during the COVID-19 pandemic may be a reflection of the complexity and rapid nature of the required public health response to the pandemic, but we believe it may also be a symptom of historic lines of jurisdictional responsibility and low levels of priority and awareness of child protection concerns. 

There has been an obvious and problematic lack of proactive early planning for child protection in the context of COVID-19 and many missed opportunities to develop child maltreatment mitigation initiatives. Black, Indigenous and low-income families have disproportionately experienced the greatest negative consequences of the COVID-19 pandemic [2]. At the same time, these communities have reported mistrust of the child welfare service due to experiences of racism and colonial policies associated with child protection in Canada [38]. Potential mitigation strategies in this context are thus complicated. Prior to the pandemic, the Ontario Association of Children’s Aid Societies [39] identified research priority areas including the provision of culturally appropriate services (for Indigenous, newcomer and diverse ethnocultural populations in Ontario), understanding and addressing child maltreatment in the foster care system; understanding and working to address limitations to child maltreatment policy and coverage for older adolescents. Urgent research is now needed to address the nature, prevalence and distribution of negative child maltreatment outcomes associated with the pandemic and public health countermeasures in Canada, as well as development and evaluation of interventions for effective child protection for all children going forward. In July 2020, the Hospital for Sick Children in Toronto provided school re-opening guidance stating that, “The community based public health measures implemented to mitigate COVID-19 and “flatten the curve” have significant adverse health and welfare consequences for children.” [11], p. 2. We believe these patterns could have been better foreseen and more centrally considered in pandemic plans. Child maltreatment is a pressing concern at all times, but even more so now as we face COVID-19 and all of its implications. Table 2 outlines possible actions that can be taken across multiple groups of Canadian stakeholders to support child protection in the current context [40,41,42,43,44].

We must recognize child protection as relevant to us all and move swiftly to ensure better mitigation and support structures here in Canada, and beyond. This is not just a concern during the COVID-19 pandemic, but exists as a concerning pattern that should be corrected more broadly. An international comparison of responses during the COVID-19 pandemic and further in-depth analysis of the strengths and weaknesses of the Canadian child protection system would be important next steps.

## Figures and Tables

**Figure 1 ijerph-18-06851-f001:**
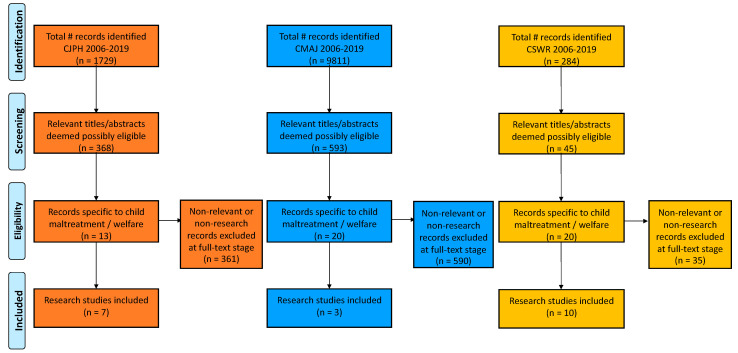
Flow diagram of the article review for the Canadian Journal of Public Heath (CJPH) (orange), the Canadian Medical Association Journal (CMAJ) (blue) and the Canadian Social Work Review (CSWR) (yellow) from 2006 to 2019.

**Table 1 ijerph-18-06851-t001:** Information about the CJPH, CMAJ and CSWR research articles specifically pertaining to the Canadian child welfare system or child maltreatment from 2006 to 2019.

**Canadian Journal of Public Health**		
**Citation Number and Title**	**Overview**	**Sample**
[15] Prévalence et co-occurrence de la violence envers les enfants dans la population Québécoise/Prevalence and co-occurrence of violence against children in the Quebec population.	Study of the prevalence and co-occurrence of various forms of violence against children in Quebec (physical, sexual and psychological). Telephone survey.	N = 822 adults asked about their childhood experiences.
[16] The Cedar Project: Negative health outcomes associated with involvement in the child welfare system among young Indigenous people who use injection and non-injection drugs in two Canadian cities.	A cohort of Indigenous Peoples aged 14–30 who use illicit drugs in Vancouver and Prince George, British Columbia. Authors explored associations between involvement in the child welfare system and vulnerability to HIV infection.	N = 605 participants, 65% had been taken from their biological parents.
[17] Québec Incidence Study on the situations investigated by child protective services: Major findings for 2008 and comparison with 1998.	Secondary analysis of data from the Québec Incidence Study on situations investigated by child protective services in collaboration with all 16 Québec child protection agencies. This study compares data from 1998 and 2008.	Two samples: 1998 (N = 4771) and 2008 (N = 3079), Quebec children investigated by child protection services.
[18] Relations spatiales entre les caractéristiques des territoires et les taux d’enfants de groupes ethnoculturels signalés à la protection de la jeunesse/Spatial relationships between the characteristics of the territories and the rates of children from ethnocultural groups reported to youth protection.	Study to map the geographic distribution of rates of children reported to Montreal child protective services by ethnocultural group (Black, other visible minorities, not from visible minorities).	N = 505, Montreal area census tracts.
[19] Prevalence, co-occurrence and decennial trends of family violence toward children in the general population.	Descriptive study using data from three large-scale telephone surveys in 1999, 2004 and 2012, to determine prevalence of psychological aggression, and minor and severe physical violence toward children.	N = 9646 mothers with children 6 months to 18 years in Quebec.
[20] Child Maltreatment and Adult Multimorbidity: Results from the Canadian Community Health Survey.	Cross-sectional, population-based study using data from the Canadian Community Health Survey to determine associations between childhood exposure to intimate partner violence, sexual abuse or physical abuse and adult multimorbidity (chronic physical conditions, pain conditions, and mental disorders).	N = 23,846 respondents aged 18+.
[21] Vaccine coverage of children in care of the child welfare system.	Retrospective cohort study of child vaccine coverage using population-based administrative health data for a 2008 birth cohort of children from Alberta, Canada. Coverage was compared for children in and not in care.	N = 44,206 at age 2 years; N = 42,241 at age 7 years.
**Canadian Medical Association Journal**		
**Citation Number and Title**	**Overview**	**Sample**
[22] Suicide and suicide attempts in children and adolescents in the child welfare system.	Population-based study examining suicide and suicide attempt outcomes for children within and outside the child welfare system in Manitoba Canada, between 1 April 1997 and 31 March 2006.	Population-level data for children 5–17 years. N = 8279 children in care; 353,050 children not in care.
[23] Child abuse and mental disorders in Canada.	Prevalence study using nationally representative data from the Canadian Community Health Survey 2012. Outcomes included experiences of child abuse, mental health conditions, suicide and suicide ideation.	N = 23,395 adults who were asked about experiences of child abuse before the age of 18 years.
[24] Prenatal Care Among Mothers Involved With Child Protection Services in Manitoba: A retrospective cohort study.	Population-based cohort of women whose first two children were born in Manitoba, Canada, between 1 April 1998, and 1 March 2015. Compared level of prenatal care between mothers with/without a child placed in care of child protection services.	N = 52,438 mothers; N = 1284 mothers with child in care.
**Canadian Social Work Review**		
**Citation Number and Title**	**Overview**	**Sample**
[25] On the Tightrope: Making sense of neglect in everyday child welfare practice. 2011; 28(2): 173–188.	Qualitative study of child neglect as understood by experienced child welfare practitioners from British Columbia, Canada.	N = 7
[26] La questions des abus sexuels en sport: Perceptions et réalité/The issue of sexual abuse in sport: Perceptions and reality.	Qualitative study using interviews and analysis of written documents to determine the nature and extent of sexual abuse prevention and child protection activities in sport organizations in Quebec.	N = 27 sport organizations; N = 6 parents; N = 9 child athletes; N = 5 coaches.
[27] Occurrence unique et concomitance de l’agression psychologique et de la punition corporelle envers les enfants/Single and concomitant occurrence of psychological abuse and corporal punishment against children.	Telephone survey to document prevalence and progression, over a five year span, of psychological aggression and physical punishment used on children.	N = 3148 mothers with children under 18 years.
[28] L’état de stress-posttraumatique-complexe et les pratiques educatives de mères d’enfants victims d’agression sexuelle: Étude de leur relation avec les symptômes des enfants/Complex Posttraumatic Stress Disorder and Educational Practices of Mothers of Sexually Assaulted Children: Examining Their Relationship to Childhood Symptoms.	Cross-sectional data from self-report questionnaires used to study the relationship between mothers’ symptoms of complex post-traumatic stress disorder, child psychological symptoms and experiences of sexual abuse.	N = 96 mothers with children aged 6–12.
[29] Knucwénte-kuc re Stsmémelt.s-kuc Trauma-informed Education for Indigenous Children in Foster Care.	Community-based research project on unceded Secwepemc territories in British Columbia, Canada. Authors use trauma-informed education with Secwepemc children, and other Indigenous children in foster care in Secwepemc Territory.	N = 40 participants.
[30] Le placement auprès de personnes significatives au Québec: Portrait des enfants placés et du contexte d’intervention/Placement with significant people in Quebec: Portrait of children in care and the context of intervention.	Cohort study of children evaluated by child protection services in 16 centres in Quebec between September 2007 and November 2009. Descriptive information about their well-being and experiences are compared between children placed in foster care and those cared for by a designate “significant person”.	N = 941 children placed with a “significant person”; N = 1586 children placed in a foster home.
[31] “Act like my friend” Mothers’ recommendations to improve relationships with their Canadian child welfare workers.	Qualitative study of abused women’s experiences in the child welfare system. Women were from communities in northern and southern British Columbia, and a larger urban centre in Manitoba.	N = 64.
[32] Liens entre le roulement du personnel vécu et l’évolution clinique d’adolescentes/Links between experienced staff turnover and the clinical development of adolescent girls.	Study of the association between the rate of staff turn-over and the experiences of female adolescents in a residential rehabilitation unit within the child protection service system.	N = 157 adolescents.
[33] Profil psychosocial des enfants présentant des comportements sexuels problématiques dans les services québecécois de protection de l’enfance/Psychosocial profile of children with problematic sexual behavior in Quebec child protection services.	Secondary analysis of data from the Quebec Incidence Study exploring the psychosocial profile of adolescents in the child protection system with and without problematic sexual behaviour (PSB).	N = 72 adolescents with PSB; N = 948 adolescents without PSB.
[34] Trajectoires de Services des Jeunes sous la Double Autorité de la Protection de la Jeunesse et de la Justice Juvénile: Différences et spécificités/Trajectories of Youth Served Under the Dual authority of Youth Protection and Juvenile Justice: Differences and specificities.	Descriptive, comparative analysis of the trajectories of youth simultaneously involved in both the child protection system and the youth criminal justice system in Quebec.	N = 15,851 youth.

**Table 2 ijerph-18-06851-t002:** Examples of Possible Mitigation and Support Structures for Child Protection.

**Governments**
Provide leadership and raise the urgency of the issue
Provide funding for immediate term initiatives
Facilitate stakeholder coordination
Support the development of non-discriminatory child protection systems and data collection in the longer term.
Provide leadership and raise the urgency of the issue
**Public health organizations**
Continue to work with other stakeholders to coordinate and implement interventions
Launch additional public education and awareness campaigns
Leverage existing data collection for evidence based decision-making for child protection.
**Healthcare providers**
Watch for signs of possible child maltreatment when dealing with children and families
Consider additional training in trauma-informed care
Take both a short and long term view when considering the ramifications of COVID-19 on child and family welfare
**Child & family organizations/advocates**
Continue engaging with clients in non-discriminatory, culturally appropriate and reassuring ways
Speak up about concerns and the need for further child protection intervention and resources
Add targeted interventions including virtual and in-person victim and survivor outreach where it is not yet provided
Expand helpline services and safe accommodation spaces
**Researchers**
Use available evidence to analyze this issue and raise awareness of child protection concerns especially related to COVID-19
Advocate for better data around child protection issues
Provide your expertise to support evidence-based decision-making
Consider adding studies of child protection issues to your own program of research
**Teachers**
Continue to watch for possible signs of child maltreatment in ongoing interactions with students
Be familiar with new resources to support teachers in their child protection role during COVID-19
**Community members**
Engage in small acts of support across our neighbourhoods and communities
Make use of technology to check in with families who have children
Share helpline information and other resources on social media, in your neighbourhoods and social circles
Look for signs of distress in others and know how to respond
Be non-judgmental and supportive when offering of information to others
**Family members**
Reach out to local, regional or national help lines if you are in need of immediate help
In this stressful time, familiarize yourself with helpful coping strategies and resources
Contact health care providers for information on parent and child supports, domestic violence, and safe spaces.
Call a close friend, relative, faith leader or community agency you trust to discuss any concerns you might have.
**Children and adolescents**
Reach out to local, regional or national helplines if you are in need of immediate help
If you are still in contact with your teacher, ask them for help
Keep an eye and ear open for signs of abuse among your peers and classmates and tell a trusted adult about what you notice.
Advocate where you can for more attention towards child protection issues
**Other community organizations**
Coordinate with faith-based organizations, educational institutions, mental health service providers and others to ensure regular virtual or in-person check-ins with higher risk families
Development of parent or family support groups to build new social support structures in neighbourhoods and communities
Speak up about the need for expanded child protection services

## Data Availability

Full results from the journal literature searches are available upon request to the corresponding author.

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
