# Peer review of "Child Maltreatment and Public Health: Do Gaps in Response during the COVID-19 Pandemic Highlight Jurisdictional Complexities?"

_ijerph, 2021, doi:10.3390/ijerph18136851_

Round 1
Reviewer 1 Report
Title: Very good title as a commentary paper!
Abstract: Very long abstract! Lines 10-15 seem not relevant here. Amid ravaging COVID-19 pandemic “the aim of this exploratory…” (lines 15-18) may be enough as a background in the abstract section. Methods, synthesis, discussion and not conclusion (line 26 vs line 132) can be slanted for emphatic demarcation. This approach may make it easily readable. NB: You can conclude in your discussion also. In addition, why did the authors used “Synthesis” (line 23) and “Result” in line 111?
Keywords: Perhaps child protection can be added to the keywords.
Introduction: The introduction section seems very okay. However, a previous study on the topic is recommended below (study 2) on the limitation and relativity of the concept of child maltreatment. Perhaps, this can further enrich this section and the discussion section too!
Considering that the nation of Canada has always being a global referent as a very good example of a multicultural society, while not support your findings on the issue of child maltreatment with ethnocultural diversity perspectives as in both studies suggested below (1 & 2) and with multidisciplinary approaches (systems thinking) in study 2 to child protection?
- Bernard, C.; Gupta, A. Black African Children and the Child Protection System. Br. J. Soc. 2008, 38, 476–492.
- Akintayo, T. Options for Africa’s Child Welfare Systems from Nigeria’s Unsustainable Multicultural Models. Sustainability 2021, 13, 1118. https://doi.org/10.3390/su13031118
Methods: In my own opinion, the method for achieving the aim of the commentary paper is seemingly very adequate and well executed.
Result: The result is seemingly well presented.
Discussion: The discussion was well focused on the result and the aim of the research. I think there is a good marriage of the title with the result and discussion sections.
References: The authors and editors are to agree if the referencing style used in this commentary paper is accepted to MDPI or the journal in particular.
Author Response
Thank you very much for your review. I have documented our revisions based on your comments.
- We have shortened our abstract based on your suggestions, thank you.
- We have added child protection as a key word.
- We have adjusted the abstract wording to be "results" versus "synthesis" and "discussion", we have also adjusted the italics as you suggest.
- Thank you for the two reference suggestions, we have honed the focus of our commentary and have not been able to include them. However, we have added an explicit statement indicating that an international comparison and further in-depth analysis of the strengths and weaknesses of the Canadian child protection system would be an important next step. Thank you.
Reviewer 2 Report
Thank you for the opportunity to review this paper. The topic is timely and important, and work in this area has the potential to improve the public health approach to child maltreatment. However, I have a number of concerns about this paper that I would like to see addressed before it is suitable for publication.
Background:
- The authors provide some background highlighting the need of child maltreatment mitigation plans in the context of the COVID-19 pandemic. This is nicely framed, and appropriately outlines the multiple ways in which children who are at-risk of being maltreated have experienced increased risk and decreased protection during COVID.
- The authors indicate that Canada in particular has not engaged in a coordinated mitigation response, leaving children in Canada particularly at risk. Obviously, it is quite impossible to demonstrate the lack of something, but it would bolster this point if comparisons to other countries/regions who did engage in a coordinated mitigation response were made. What might this response have looked like, if it were carried out well? In order to demonstrate that the failure to respond during COVID is linked with underlying public health issues, it would be helpful to know how other areas have done well in this realm, despite dealing with the COVID pandemic.
Method
- More methodological details could be provided in this section.
- What about some report of demographics of the participant from these selected studies? This is especially important considering concerns that Canadian child maltreatment research may not include diverse populations compared to other regions, as noted in the introduction.
- What were the research questions asked by the included studies?
- How did authors ensure that publications utilizing the same dataset or project were not included, or if they were, how this was handled?
- Were articles screened to ensure they represented Canadian child maltreatment samples, and not samples from other countries that were published in this article?
- Overall, it is not clear what the aim of reviewing these articles was. The authors state that their aim was to examine the extent and nature of child maltreatment research, but simply reporting the number of studies included does not accomplish this goal.
Results
- The results suggest that there were no obvious pattern of differences or similarity in the focus on research designs or topics. This sentence itself seems contradictory – they were not similar or different from one another?
- Additionally, without knowing the qualitative review process conducted, the reader cannot know about the ways in which these articles were screened for similarity and difference.
Discussion
- The points made in this discussion are important ones, and a good commentary on the ways in which the child welfare system and COVID have intersected to disproportionately negatively impact BIPOC families, and children generally. However, this discussion seems largely disconnected from the method and results of this paper.
Author Response
Thank you for your review of our manuscript, we have noted below our responses to your comments:
- It would bolster this point if comparisons to other countries/regions who did engage in a coordinated mitigation response were made. What might this response have looked like, if it were carried out well?
We agree with this comment, unfortunately as far as we can find, there has been a real lack of documented examples of instances where this was done well. There have been many more calls of concern internationally. We have added a more obvious focus on the within Canada context in the abstract and introduction and have noted that an international comparison and in-depth, international analysis of child protection in light of COVID-19 responses is a ideal next step. We have added examples of ideal mitigation and response ideas in Table 2.
2. What about some report of demographics of the participant from these selected studies?
We have now added details about each study sample in Table 1.
3. The results suggest that there were no obvious pattern of differences or similarity in the focus on research designs or topics. This sentence itself seems contradictory.
We have adjusted the language in this sentence to be more clear.
4. What were the research questions asked by the included studies?
We have now added details about each included study in Table 1.
5. How did authors ensure that publications utilizing the same dataset or project were not included, or if they were, how this was handled?
Details of the datasets used are now included in Table 1 so that this is transparent for the reader.
6. Were articles screened to ensure they represented Canadian child maltreatment samples, and not samples from other countries that were published in this article?
Details about the samples for each study are now included in Table 1. These did not include international samples.
7. Overall, it is not clear what the aim of reviewing these articles was.
Thank you for this comment, we have restated our aim in the abstract and discussion so that it is consistent and more clear.
8. This discussion seems largely disconnected from the method and results of this paper.
We have adjusted the first and last paragraphs of the discussion so that it is clear what the implications of the results are. We did not choose to delve into the details of each of the study contents, but rather, as part of our commentary noted the overall lack of academic literature (despite discipline) and the real need for greater emphasis in this area.
Reviewer 3 Report
I think this is a nice little study making for a nice little commentary. These kinds of articles are always useful to have, so I would support its publication on the basis of that alone. It is well-written, and the methods and findings clearly presented. I was also pleased that the authors did not overly ‘quantify’ the findings. Far too often I find that authors of literature reviews (which this is) insist upon presenting the findings in an over-wrought and unnecessarily complicated empirical/quantitative framework. Yes, there is a requirement for some empirical findings here in order to make the argument, but the authors did not get carried away with this and did not lose sight of the qualitative: what these articles were about.
I do have a few comments for the authors; these are not necessarily demands for change though. The underlying rationale is sound – I am also disturbed (and worried) what we will find once COVID runs its course, particularly regarding the long-term impacts on children and youth. The authors also point out an issue that vexes me – why there is not more co-operation among responsible parties (though in my country part of the reason for this lack of co-operation is due to the local child protection committees themselves – but that is another issue). However, while I understand the selection of a social work and public health journal, to be frank child protection is my area of research as well and I don’t bother looking at medical journals unless I am looking for something very specific. I don’t know the Canadian context well – though I work with a half dozen or so Canadian colleagues in this field – but I would probably turn to non-medical journals for this kind of evidence, maybe the Canadian Journal of Family and Youth or sources like that. As I said, I do not know the context like the authors, but a finding of low references to child maltreatment in CMAJ does not really surprise me – but it does with regard to the social work journal. However, point taken and maybe they should focus on this more.
I am a little more dubious though of the claim on page 5: “This represents a potentially limited evidence base from which to inform policy and decision making.” Well, I am not sure I agree. Again, I don’t know the Canadian context, but where I conduct research the government as well as child protection committees do not tend to pay much attention to academic sources. The evidence they tend to pay attention to are the streams of reports from local child protection committees, the national child protection agency, our version of the Surgeon-General, directorate of health reports and so forth. I agree the authors make the case that there is limited academic evidence, but I am not sure about the extent to which there is limited evidence overall in regard to the output of national, provincial and municipal agencies. It is a bit sobering as a fellow academic, but academic evidence in my experience does not seem to have as much impact on the work of public authorities as the authors appear to assume. Maybe Canada is different. And maybe the authors have another area of inquiry as to whether or not these agencies are looking into COVID impacts as well. It might help to amend this sentence with limited ‘academic’ evidence if there are indeed other forms available, e.g. reports from public agencies etc.
Author Response
We greatly appreciate your review of our manuscript and have documented our revisions based on your comments.
1. A finding of low references to child maltreatment in CMAJ does not really surprise me – but it does with regard to the social work journal...maybe they should focus on this more.
We have added a sentence stating that the very limited literature base was not expected, especially is a social work journal.
2. I am a little more dubious though of the claim on page 5: “This represents a potentially limited evidence base from which to inform policy and decision making.” It might help to amend this sentence with limited ‘academic’ evidence.
Thank you for this suggestion and we agree. We have adjusted and added "academic" in many places to ensure the type of evidence we are talking about is clear.
Round 2
Reviewer 2 Report
I appreciate the authors' responsiveness to the concerns that I raised during the first review. I think that this version clearly communicates that this paper is a review of existing empirical studies of the child welfare system prior to COVID-19, and that the clear conclusion is that very few researchers are publishing in this area (at least by review of the selected journals). The authors claim that perhaps this is due to a lack of "jurisdiction" across disciplines with regard to child welfare.
I would urge the authors to reconsider the title of this paper, and to choose one that better fits the aims and conclusions of this paper. As it is currently titled, it suggests that a lack of preparation to COVID-19 was driven by "broader concerns" but I think a more direct title would relate to the main points of the authors (lack of "jurisdiction", and overall lack of empirical child welfare research leading to underpreparedness for COVID-19, etc.)
I would also suggest that the Noel article listed under the social work journal be reconsidered. I don't know that this is an empirical article, as much as a theoretical paper.
Author Response
Thank you very much for you follow-up comments. We have adjusted the title based on your suggestion and have removed the Noel citation for consistency. We have adjusted the results figure and discussion text to ensure they align with this change in numbers. We have also added the word "jurisdiction" twice in the discussion to align better with the commentary title as well.
